# The Interplay between Mitochondrial Metabolism and Nasal Mucociliary Function as a Surrogate Method to Diagnose Thyroid Dysfunction: Insights from a Population-Based Study

**DOI:** 10.3390/biomedicines12081897

**Published:** 2024-08-20

**Authors:** Mohammad Farhadi, Hadi Ghanbari, Ali Salehi, Sumel Ashique, Farzad Taghizadeh-Hesary

**Affiliations:** 1ENT and Head and Neck Research Center and Department, The Five Senses Health Institute, School of Medicine, Iran University of Medical Sciences, Tehran 14496-14535, Iran; farhadi.m@iums.ac.ir (M.F.); ghanbari_md@iums.ac.ir (H.G.); dr.ali_salehi89@yahoo.com (A.S.); 2School of Pharmaceutical Sciences, Lovely Professional University, Phagwara 144411, Punjab, India; ashiquesumel007@gmail.com; 3Department of Pharmacy, Bharat Institute of Technology (BIT), School of Pharmacy, Meerut 250103, Uttar Pradesh, India

**Keywords:** cilia, mitochondria, nasal mucociliary clearance, thyroid dysfunction, thyroid-stimulating hormone

## Abstract

**Aim and Background**. This study aims to explore alternative diagnostic methods to assess thyroid function in patients unable to undergo blood tests for thyroid-stimulating hormones (TSH) and thyroxine (T4), such as individuals with trypanophobia, severe medical conditions, or coagulopathy. Considering the impact of thyroid dysfunction on mitochondrial metabolism and the essential role of proper mitochondrial function in ciliary motility, we postulate that assessing nasal ciliary function could serve as a surrogate diagnostic approach for thyroid dysfunction. **Methods**. This cross-sectional study was performed on individuals with no history of thyroid diseases. The primary endpoint was the diagnostic value of the nasal mucociliary (NMC) test using Iranica Picris (Asteraceae) aqueous extract in differentiating hypo- or hyperthyroidism cases from euthyroid cases. **Results**. 232 individuals were recruited (71% females, 86% euthyroid). Receiver operating characteristic (ROC) analysis showed a good diagnostic value for the NMC test in differentiating overt hypothyroidism (area under the ROC curve [AUROC] = 0.82, *p* = 0.004) and its fair value in diagnosing subclinical hyperthyroidism (AUROC = 0.78, *p* = 0.01) from the euthyroid condition. The NMC test had a significant positive correlation with TSH (*r* = 0.47, *p* < 0.001) and a significant negative correlation with T4 (*r* = −0.32, *p* < 0.001). The NMC rate was significantly different in distinct thyroid function groups (*p* < 0.001). Compared with euthyroid cases, the post-hoc analysis showed that the NMC test is significantly higher in overt hypothyroidism (15.06 vs. 21.07 min, *p* = 0.003) and significantly lower in subclinical hyperthyroidism (15.05 vs. 10.9 min, *p* = 0.02). **Conclusions**. The Iranica Picris-based NMC test might serve as a diagnostic method to distinguish overt hypothyroidism and subclinical hyperthyroidism.

## 1. Introduction

The thyroid gland plays a vital role in human physiology by regulating metabolism and facilitating proper body development and function. Undiagnosed thyroid disorder can pose a significant risk to hosts, potentially leading to serious conditions such as cardiovascular diseases, osteoporosis, and infertility. According to the American Thyroid Association (ATA), over 12% of the United States’ inhabitants experience a thyroid condition during their lives. It is believed that around 20 million Americans are currently living with a form of thyroid disease, but that, shockingly, up to 60% of those affected are unaware of their condition [1]. Therefore, early detection of thyroid dysfunction is of crucial importance. Current practice for diagnosing thyroid dysfunction is primarily based on laboratory tests.

The thyroid-stimulating hormone, or thyrotropin, (TSH) is a pituitary hormone, physiologically secreted in response to a reduction in thyroid hormones. Given its amplified response to slight changes in thyroid hormone levels (i.e., T3 [triiodothyronine] and T4 [thyroxine]), TSH has been widely examined in clinical practice to detect subtle drops in thyroid function. Nowadays, the most common panel used to differentiate thyroid dysfunction is the combination of TSH and total T4 [2]. However, the interpretation of this panel might be complicated due to the multiple endocrine glands and hormones involved in its negative feedback loop, as well as the various tissues, hormones, and biochemical pathways involved in overall thyroid function. In addition, several factors can interfere with the TSH response to serum thyroid levels, including demographic (age, gender, ethnicity) and environmental (smoking, medicines) factors, concomitant physiologic conditions (pregnancy), and diseases (pituitary adenoma, autoimmune diseases). Even the time of blood sampling can impact the TSH level [3]. Although serum TSH and T4 levels are considered the gold standard for diagnosing thyroid dysfunction, individuals with *trypanophobia*, or fear of needles, may encounter challenges in undergoing blood tests to assess their thyroid function. Furthermore, individuals with severe medical conditions and coagulopathy face limitations in diagnosing thyroid function through blood samples. In such cases, there is currently no established method to distinguish thyroid function status without obtaining blood samples. These limitations urged us to find an alternative method to diagnose thyroid function disorders. 

Thyroid hormones can boost energy production in mitochondria by promoting mitochondrial replication and shifting metabolism from glycolysis to the more efficient oxidative phosphorylation [4]. Mitochondria are essential organelles responsible for providing energy (in the form of adenosine triphosphate, ATP) for various functions in different tissues of the human body [5,6], including the cilia in the nasal epithelium [7]. Nasal cilia are small, slender, hair-like structures lining the nasal epithelium. Benefiting from rhythmic beats (7–16 Hz), cilia can remove particles—trapped in the gelatinous mucosa—from the airways. The nasal mucus path ends in the pharynx, where it is eventually swallowed. This process is called nasal mucociliary clearance (NMC), and is a component of the innate immune system [8]. 

The most available method to evaluate the NMC rate is via saccharine test. In this test, a saccharine tablet is placed adjacent to the inferior concha, and the time required to sense a sweet taste is recorded as the NMC rate [9]. The NMC rate can be influenced by physiologic factors, such as age, gender, posture, sleep, exercise, and body temperature [10]. Pandya et al. demonstrated that the mean NMC rates for children and adults are 11.1 and 12.7 mm/min, respectively [11]. It has been shown that hypothyroidism can lead to mucociliary dysfunction. Uysal et al. evaluated the effects of hypothyroidism on nasal mucociliary clearance in patients undergoing total or near total thyroidectomy who planned to receive radioactive iodine. The study demonstrated that mucociliary clearance time was significantly longer during hypothyroid periods as compared to euthyroid periods (16.7 min vs. 9.5 min) [12].

Given this background, we hypothesized that the NMC rate can address the thyroid function status. The findings of this study can serve to introduce a novel diagnostic test for thyroid dysfunction. In this study, we applied a new method to measure the NMC rate. Given the path of nasal mucus from the nasal cavity to the pharynx, substituting saccharine (sweet taste) with a bitter substance (which is greatly perceived in the posterior tongue) can improve the sensitivity of the NMC test [13,14]. As such, Iranica Picris (Asteraceae) aqueous extract (IPAE) was chosen to test the NMC rate. To test our hypothesis, we conducted a population-based study to determine the correlation between NMC rate and thyroid function tests and its diagnostic value for differentiating thyroid function disorders.

## 2. Materials and Methods

### 2.1. Study Design, Participants, and Endpoints

This is a cross-sectional, population-based study of individuals with no past medical history of thyroid disease. Participants were referred to the Otorhinolaryngology Department of Rasoul Akram Hospital (Tehran, Iran) for purposes other than thyroid care. Routine ear, nose, and throat examination was performed in the otorhinolaryngology clinic. The exclusion criteria were a history of chronic rhinosinusitis with polyps or any other anatomic abnormalities in the imaging (e.g., marked septal deviation) impeding the NMC test procedure, and the consuming of medicines that interfere with TSH levels (including corticosteroids, dopaminergic agents, amiodarone, contraceptive agents, etc.). The primary endpoint of this study was to evaluate the diagnostic value of IPAE-based NMC to differentiate patients with thyroid dysfunction from euthyroid individuals. The secondary endpoints were the correlation of NMC time (NMCT, in minutes) with both TSH and T4 changes and NMCT changes over the study groups—per TSH and T4 levels and thyroid function (i.e., hypo-, hyper-, and euthyroid)—and to calculate the sensitivity and specificity of the NMC test in detecting thyroid dysfunction. 

### 2.2. Sample Size Calculation

To make the results representative of the general population, before the study recruitment, the minimum sample size was calculated using the following formula [15]:(1)$$z^{2}\times p(1-p)/\varepsilon^{2}$$
where z is the Z score, p is the population portion, and ε is the margin of error. Given the confidence interval of 95% (CI 95%), the margin of error of 5%, and the population portion of 15%, the minimum required number of participants was 196. The population portion was defined based on the available studies; a population-based study from Iran found that the overall prevalence of thyroid function disorders (sum of overt hypo-, subclinical hypo-, subclinical hyper-, and overt hyperthyroidism) was 15.7% [16]. 

### 2.3. Assessments

To evaluate the NMCT, cotton soaked in IPAE was applied to the inferior turbinate surface, about one cm behind the membranous columella. To minimize bias, the NMC rate was tested at room temperature for the upright sitting posture. In addition, participants were examined for body temperature to exclude a fever of more than 38 °C. The participants were instructed to clear their noses and refrain from eating and drinking one hour before the test. They were also asked to withhold from sniffing, coughing, sneezing, or consuming any food or beverage during the procedure and evaluation. With proper mucociliary action, the IPAE could be swept back to the pharynx. The time required to perceive the bitter taste of IPAE was registered in minutes. The step was repeated for the contralateral nostril. The mean clearance time was then calculated as the average NMCT of the right and left nostrils. Next, the individuals were sent to the laboratory department to assess their serum TSH and total T4 levels. It has been demonstrated that this combination is the best-suited panel to assess hypo- or hyperthyroidism [2].

### 2.4. Ethical Issues

Participants (and their parents for younger-aged individuals) were educated about the procedure process, its safety, and its implications, and filled out the informed consent before the procedure. The experimental protocols followed the established guidelines and regulations and were approved by the Institutional Review Board of Iran University of Medical Sciences (Tehran, Iran). The study adhered to the principles of the Declaration of Helsinki. As human evidence on the safety of IPAE was available [17] and the NMC test is inherently harmless, additional ethical clearance was waived by the Institutional Review Board. The reporting of this cross-sectional study follows the STROBE checklist (available at: https://www.strobe-statement.org/checklists, accessed on 10 July 2023).

### 2.5. Statistical Analysis

Categorical variables were summarized as numbers and percentages, and continuous variables were summarized as mean, range, and standard deviation. The Shapiro-Wilk test was applied to test the normal distribution of variables, and demonstrated that TSH, T4, and NMCT values do not have a normal distribution. As such, the Spearman’s Rank test was used to determine the correlation of NMCT with TSH and T4. The strength of the correlation coefficient values was rated based on Chan’s study as follows: [values] ≥ 0.8 (very strong), 0.6–0.8 (moderately strong), 0.3–0.5 (fair), <0.3 (weak) [18]. The Kruskal-Wallis test was applied to determine any difference in NMCT between the different TSH, T4, and thyroid function categories. Next, the corresponding post-hoc analysis was run using Dunn’s test. The receiver operating characteristic (ROC) curve analysis was used to determine the diagnostic value of the NMC test in differentiating different thyroid function disorders. The results of the area under the ROC curve (AUROC) were determined to be excellent when the AUC values were between 0.9 and 1.0, good for values between 0.8 and 0.9, fair for values between 0.7 and 0.8, poor for values between 0.6 and 0.7, and failed for values between 0.5 and 0.6 [19]. Sensitivity and specificity values for the Iranica Picris-based NMC test were calculated based on the maximum likelihood ratio (Sensitivity/(1−Specificity)), as determined from the data obtained in the ROC curve analysis. The GraphPad Prism 9.5.1 (Dotmatics, San Diego, CA, USA) was applied for statistical analysis, and the statistical significance was set to 0.05. 

## 3. Results

### 3.1. Descriptive Analysis

Table 1 indicates the distribution of data based on demographic, thyroid function tests (TFTs), and the NMCT of study participants. A total of 232 patients were included, of whom 165 (71.1%) were female, and of whom 150 (64.6%) were 50 years old or younger (median age, 44 years; range, 10–82). Most participants had a normal range of TSH (86.6%) and T4 (95.3%). According to the TFT results, 201 participants (86.6%) were categorized as euthyroid. Cases with subclinical hypo- or hyperthyroidism did not have clinical manifestations of thyroid dysfunction. The raw data of NMCT values of the participants are outlined in Table 1. The mean and median values of average NMCT were 15.2 and 15 min, respectively. The comparison of right and left NMC test values showed no significant difference (mean 15.3 vs. 15.1, *p* = 0.49). Figure 1 demonstrates the distribution of TSH, T4, and NMCT values in the study population. It shows that TSH has a multimodal distribution, T4 has a unimodal, right-skewed distribution (skewness value = +2.8), and NMCT has a relatively uniform distribution. NMC procedures had no adverse effect per the patients’ reports and clinical examination. 

### 3.2. Correlation between Nasal Mucociliary Clearance Time and Thyroid Function Tests

Next, the correlation between NMCT and TFTs was evaluated. A Spearman’s rank-order test was run to this end. There is a fair, positive correlation between NMCT and basal TSH concentration, which is statistically significant (*r* = 0.47, *p* < 0.001) (Figure 2A). The correlation between NMCT and T4 values is also at the ‘fair’ level and statistically significant but inverse (*r* = −0.32, *p* < 0.001) (Figure 2B). For interpretation of the correlation coefficient, please refer to the Section 2.5. The partial correlation analyses showed that the correlations remained significant with the TSH levels when the effects of age and gender were controlled (age: *r* = 0.32, *p* < 0.001; gender: *r* = 0.30, *p* < 0.001). Similar results were found for NMCT correlation with T4 when controlled for other factors (age: *r* = −0.25, *p* < 0.001; gender: *r* = −0.33, *p* < 0.001).

### 3.3. Mucociliary Clearance over the TSH, T4, and Thyroid Function Groups

Next, we analyzed whether there is any statistically significant difference among different groups in terms of NMCT values. The Kruskal-Wallis test determined that there are statistically significant differences among the TSH groups (H = 17.6, *p* < 0.001), T4 groups (H = 11.7, *p* = 0.002), and thyroid function groups (H = 19.7, *p* < 0.001) in terms of the NMC rate (Figure 3A–C). 

Regarding the TSH, a Dunn’s post-hoc test revealed that the NMCT values are significantly different between low- and normal-TSH groups (mean 11.1 vs. 15.06, *p* = 0.005), normal- and high-TSH groups (mean 15.06 vs. 18.5, *p* = 0.003), and low- and high-TSH groups (*p* < 0.001) (Figure 3A). Regarding T4, the post-hoc analysis showed that the NMCT values were significantly different between low- and normal-T4 groups (mean 21.07 vs. 15.09, *p* = 0.003) and low- and high-T4 groups (mean 21.07 vs. 11.3, *p* = 0.001). However, there was no statistically significant difference between the normal and high-T4 groups (*p* = 0.08) (Figure 3B). Regarding thyroid function, post-hoc analysis revealed a significant difference between NMCT values of participants with euthyroid and subclinical hyperthyroidism (15.05 vs. 10.9, *p* = 0.02), the most among all subsequent groups. In addition, NMCT values were significantly different between euthyroid and overt hypothyroidism (15.06 vs. 21.07, *p* = 0.003); however, no significant difference was recognized between euthyroid NMCT values and overt hyperthyroidism (mean 15.06 vs. 11.3, *p* = 0.08) (Figure 3C). Subanalysis per gender was unavailable because of the limited number of males with overt hypothyroidism and subclinical hyperthyroidism (one and none, respectively). Likewise, subanalysis per age grouping was not possible because almost all participants in the overt hypothyroidism and subclinical hyperthyroidism groups were within the middle-age group (85% and 83%, respectively).

### 3.4. Diagnostic Value of Mucociliary Clearance Test in Differentiating Thyroid Function Disorders

To determine the significance of the NMC test in diagnosing thyroid function disorders, ROC analysis was conducted, and the corresponding ROC curves were plotted (Figure 4). The ROC analysis indicates that the diagnostic value of the NMC test in differentiating overt hypothyroidism is significant and is in the ‘good’ range (AUROC = 0.82, CI 95% 0.64–0.99, *p* = 0.004) (Figure 4A), and its capability to identify subclinical hyperthyroidism from euthyroid cases is significant and ‘fair’ (AUROC = 0.78, CI 95% 0.69–0.87, *p* = 0.01) (Figure 4C). For interpretation of the correlation coefficient, please refer to the Section 2.5. However, the ROC analysis did not indicate significance of the NMC test in differentiating subclinical hypothyroidism and overt hyperthyroidism (*p* > 0.05) (Figure 4B,D). As noted earlier, subanalysis per gender and age was not possible. 

Next, the likelihood ratio, specificity, and sensitivity of the IPAE-based NMC test were calculated for cases with significant *p*-values in ROC analysis (overt hypo- and subclinical hyperthyroidism). The optimal likelihood ratio for overt hypothyroidism was 4.2, and was at the cutoff point of 16.7 min. At this point, the specificity and sensitivity of the NMC test were 85.7 and 61.4%, respectively. At the optimal cutoff of 13.2 min, the optimal likelihood ratio, specificity, and sensitivity for diagnosing subclinical hyperthyroidism were 2.7, 63.7, and 100%. 

## 4. Discussion

This population-based study showed that the IPAE-based NMC test could differentiate overt hypothyroidism and subclinical hyperthyroidism from a euthyroid condition. For values more than 16 min, the NMC rate had acceptable specificity (85%) and sensitivity (61%) for differentiating hypothyroidism, and for values less than 13.2 min, it had a sensitivity of 100% and a specificity of 63%. The NMC test had a significant positive correlation with TSH changes and a negative correlation with T4. The results also indicate the considerable power of the NMC test to differentiate normal-range TSH from high-or low-TSH values, normal-range T4 from low-T4 values, and patients with overt hypothyroidism or subclinical hyperthyroidism from euthyroid cases. These findings are in line with the literature. Uysal et al. demonstrated that thyroidectomy (total or near total) could significantly impair nasal mucociliary clearance [12]. This interplay may justify the higher susceptibility of patients with hypothyroidism to developing chronic rhinosinusitis [20]. 

We hypothesized that the NMC test can differentiate different groups of thyroid function. The results indicated that the NMC test could distinguish overt hypothyroidism (and subclinical hyperthyroidism) from euthyroid individuals. The small number of patients with overt hyperthyroidism might affect its statistically nonsignificant difference from euthyroid cases despite the large numerical difference (mean 15.06 vs. 11.3, *p* = 0.08) (Figure 3C). A deep dive into the biological effects of thyroid hormones on cell structures can justify these results as follows: 

Cilia beating is an ATP-dependent process. Kinesin family member 19A (Kif19a) is a motor protein that attaches to the cilia tubulin in an ATP-dependent manner [21]. Therefore, normal cilia dynamics depend on functional mitochondria generating enough ATP molecules [22]. Furthermore, elevated levels of extracellular ATP have been shown to enhance ciliary beating in ciliated epithelial cells by inducing membrane hyperpolarization through a calcium-dependent process [23,24]. Research suggests that mitochondria are the primary supplier of extracellular ATP in the vicinity of epithelial cells [25]. This paragraph has demonstrated how mitochondrial metabolism directly affects cilia dynamics by enhancing intra- and extracellular ATP contents.

Recent evidence shows that the influence of mitochondrial metabolism on proper cilia function goes beyond just the providing of the ATP molecules. Mitochondria are primary human cell organelles that scavenge the reactive oxygen species (ROS) produced during metabolism [26]. Therefore, mitochondrial dysfunction would increase the cellular ROS levels, resulting in cellular oxidative stress. It has been demonstrated that oxidative stress can impair ciliogenesis and its structure [27]. Therefore, functional mitochondria are necessary for the normal function of cilia by providing enough ATP molecules and preventing the adverse effects of ROS. Recent evidence has addressed the direct impact of TSH on mitochondrial metabolism. Wang et al. demonstrated that TSH can increase the mitochondrial ROS level without affecting ATP production. This effect is mediated by disrupting the mitochondrial respiratory chain [28]. This event may justify our finding on the diagnostic value of the NMC test in cases with subclinical hyperthyroidism (with normal T4 and reduced serum TSH levels). Low serum TSH can prevent excess ROS generation, improving the NMC rate by increasing the cilia number and maintaining their function. 

To date, several studies have been conducted to devise a method to improve the diagnosis of thyroid dysfunction. Ma et al. introduced gp91^phox^ as a novel biomarker for distinguishing subclinical hypothyroidism [29]. As the catalytic core of nicotinamide adenine dinucleotide phosphate (NADPH) oxidase, gp91^phox^ can indicate cellular oxidative stress. Nevertheless, Ma et al. did not clarify how gp91^phox^ can distinguish subclinical hypothyroidism. As noted in the previous paragraph, TSH can directly lead to mitochondrial oxidative stress. As such, in the setting of subclinical hypothyroidism with elevated TSH levels, oxidative biomarkers such as gp91^phox^ are elevated. It has been demonstrated that fibulin-1 is associated with thyroid-associated ophthalmopathy. In this study, Hu et al. discovered that serum fibulin-1 levels higher than 625 pg.mL^−1^ significantly increased the risk of thyroid-associated ophthalmopathy [30].

The findings of the current study must be interpreted considering the following limitations: 

(1) We evaluated the total T4 values. We suggest also considering the free (unbound) T4 in the analysis. 

(2) The population study included mainly middle-aged females. Further, larger studies are required to explore the results in males and at the extremes of age. 

(3) The impacts of tobacco smoking on TSH and T4 levels have been demonstrated [31]. The current study did not stratify the results with respect to tobacco smoking status. 

Despite these limitations, this is among a handful of studies to evaluate the impact of thyroid function on the NMC rate, and the first study to introduce the NMC test as a method of thyroid function assessment.

## 5. Conclusions

This study highlights the ability of the NMC test to distinguish overt hypothyroidism or subclinical hyperthyroidism from euthyroid individuals. TSH is a sensitive biomarker for diagnosing thyroid function disorders. However, certain conditions can diminish its application and accuracy. Given the high specificity of the IPAE-based NMC test to detect hypothyroid cases, it can serve as a complementary test to TSH, improving its diagnostic accuracy in diagnosing hypothyroidism. Notably, this test may offer a non-invasive alternative for diagnosing thyroid dysfunction in individuals with trypanophobia, those with coagulopathy, or those reluctant to undergo conventional diagnostic tests for any reason. In other words, this innovative approach could provide a valuable means for assessing thyroid function without the need of blood samples, ensuring appropriate care for such individuals. 

## Figures and Tables

**Figure 1 biomedicines-12-01897-f001:**
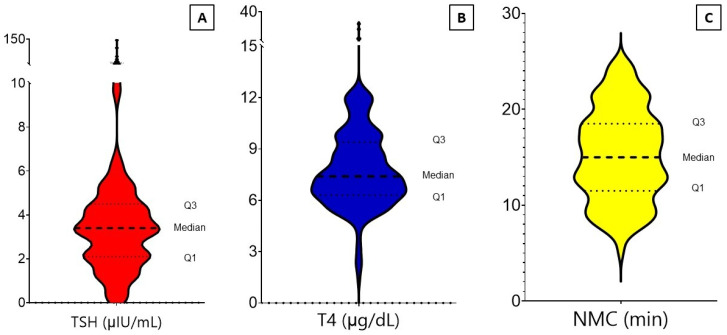
Violin plots of TSH (**A**), T4 (**B**), and nasal mucociliary test (**C**) distribution in the study population. The Q1/median/Q3 values of TSH, T4, and NMC are 2.1/3.4/4.5, 6.3/7.4/9.4, and 11.5/15.0/18.5, respectively. NMC indicates nasal mucociliary clearance time; TSH, thyroid-stimulating hormone.

**Figure 2 biomedicines-12-01897-f002:**
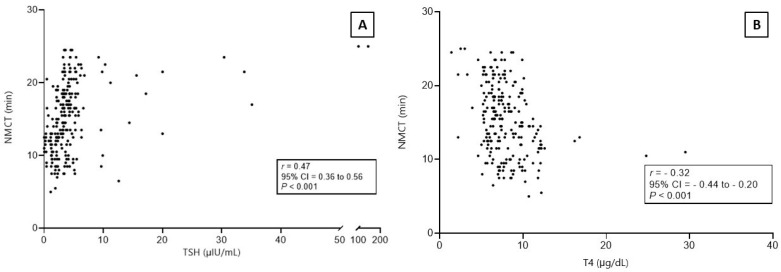
Scatterplot of TSH vs. NMCT (**A**) and T4 vs. NMCT (**B**). r indicates Spearman’s rank correlation coefficient.

**Figure 3 biomedicines-12-01897-f003:**
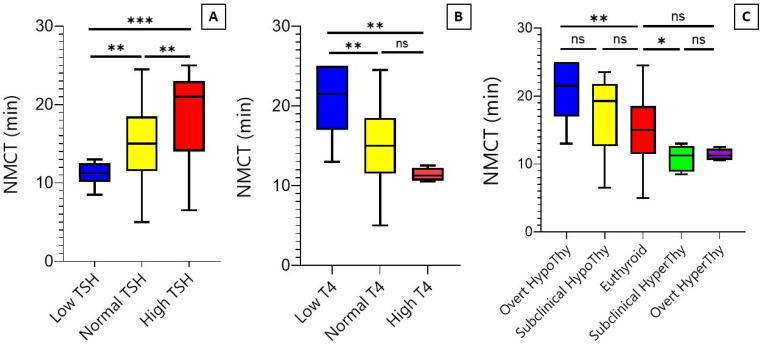
Nasal mucociliary clearance time distribution per serum TSH level (**A**), serum T4 level (**B**), and thyroid function (**C**). NMCT indicates nasal mucociliary clearance time; TSH, thyroid-stimulating hormone. *** is significant at level of 0.001, ** at level of 0.01, and * at level of 0.05, ns, nonsignificant (*p* > 0.05).

**Figure 4 biomedicines-12-01897-f004:**
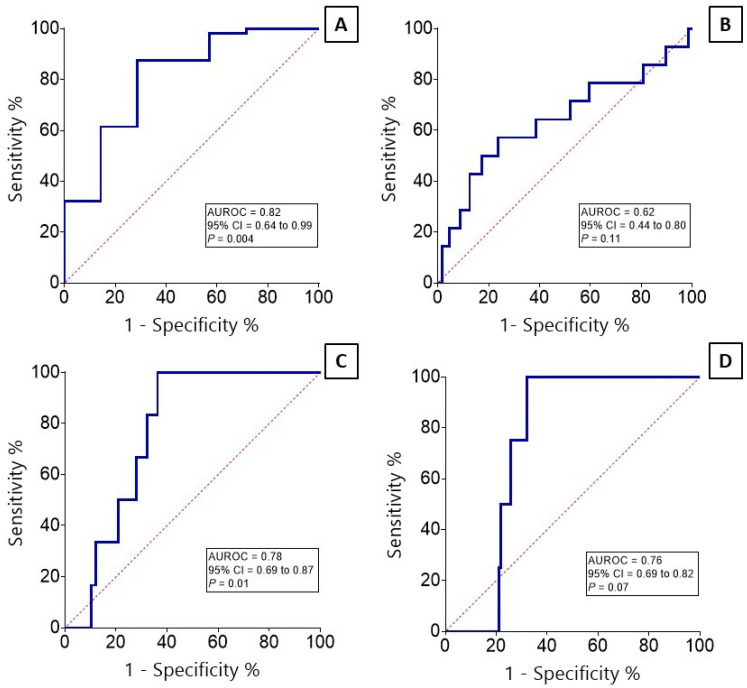
ROC curve analysis of nasal mucociliary clearance test in detecting overt hypothyroidism (**A**), subclinical hypothyroidism (**B**), subclinical hyperthyroidism (**C**), and overt hyperthyroidism (**D**). AUROC indicates area under the ROC curve.

**Table 1 biomedicines-12-01897-t001:** Basal characteristics of participants.

Characteristics	Patients (Total = 232)
Gender	
Female, *n* (%)	165 (71.1)
Male, *n* (%)	67 (28.9)
Patient age at diagnosis (years)	
10–30, *n* (%)	59 (25.4)
31–50, *n* (%)	91 (39.2)
51–70, *n* (%)	66 (28.4)
>70, *n* (%)	16 (6.8)
TSH (μIU/mL)	
Mean/range/SD	5.3/0.01–144.3/3.7
Low (<0.4)	10 (4.3)
Normal (0.4–6.1 or up to 10 in >50 years)	201 (86.6)
High (>6.1 or 10 in >50 years)	21 (9.1)
T4 (μg/dL)	
Mean/range/SD	7.9/1.4–29.5/2.8
Low (<4.5)	7 (3.0)
Normal (4.5–12.5)	221 (95.3)
High (>12.5)	4 (1.7)
Condition	
Overt hypothyroidism	7 (3.0)
Subclinical hypothyroidism	14 (6.0)
Euthyroid	201 (86.6)
Subclinical hyperthyroidism	6 (2.6)
Overt hyperthyroidism	4 (1.7)
NMCT (min)	
Right side	
Mean	15.3
Range/IQR	4–28/11–19
SD	5.10
Left side	
Mean	15.1
Range/IQR	4–31/11–19
SD	4.96
Average	
Mean	15.2
Range/IQR	5–25/11.5–18.5
SD	4.73

Abbreviations: IQR, interquartile range; NMCT, nasal mucociliary clearance time; SD, standard deviation; TSH, thyroid-stimulating hormone.

## Data Availability

The datasets supporting the conclusions of this article are available upon official request from the corresponding author.

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
