# Peer review of "The Interplay between Mitochondrial Metabolism and Nasal Mucociliary Function as a Surrogate Method to Diagnose Thyroid Dysfunction: Insights from a Population-Based Study"

_biomedicines, 2024, doi:10.3390/biomedicines12081897_

Round 1

Reviewer 1 Report

Comments and Suggestions for Authors

In this manuscript, authors proposed that nasal mucociliary function (NMC) as a method to diagnose thyroid dysfunction which is already reported widely and a well established fact. Another major concern is that there are no assays or methods performed to assess mitochondrial metabolism/function in subjects. How NMC and mitochondrial metabolism are connected? How authors even justify the title without assessing the mitochondrial metabolism?

Author Response

Dear esteemed Editor and Reviewers

We are submitting the revision of the manuscript. Thank you for your valuable comments, which have improved the quality and clarity of the paper. In the revised manuscript, the responses to the Reviewers’ comments are highlighted in yellow. We tried our best to improve the manuscript per the Reviewers’ comments. I genuinely appreciate the Editor’s and Reviewers’ valuable work and hope the corrections meet the approval requirements.

Best Regards,

Farzad Taghizadeh-Hesary, M.D. 

Esteemed Reviewer 1

Comment 1: In this manuscript, authors proposed that nasal mucociliary function (NMC) as a method to diagnose thyroid dysfunction which is already reported widely and a well established fact.

Response 1: We would like to thank the esteemed Reviewer for this feedback which was really valuable to me. We appreciate your insight into the existing literature regarding the relationship between hypothyroidism and nasal ciliary function, as evidenced by studies such as PMID: 23519683 and PMID: 32754873.

Although prior research has established that hypothyroidism affects nasal ciliary function, this investigation will be the first to propose the use of the nasal mucociliary test as a diagnostic tool to determine thyroid disorders. In addition, we realize that numerous publications have described the correlation between hypothyroidism and nasal ciliary clearance. On the other hand, our study goes further by providing evidence to show that higher levels of thyroid function might contribute positively towards ciliary clearance.

We wish this clarification has highlighted that our work offered a distinct perspective and value to the information already out there on thyroid dysfunction and nasal mucociliary function. Thanks for the attention and consideration.

Comment 2: Another major concern is that there are no assays or methods performed to assess mitochondrial metabolism/function in subjects. How NMC and mitochondrial metabolism are connected? How authors even justify the title without assessing the mitochondrial metabolism?

Response 2: First and foremost, we would like to express our profound gratitude to the esteemed Reviewer for this invaluable comment, which has significantly enriched our study.

As the esteemed Reviewer knows, experimental and clinical evidence indicates that enhancing thyroid hormone activity promotes cellular mitochondrial biogenesis and function; moreover, mitochondrial metabolism is increasingly appreciated as indispensable for keeping cilia in functional condition. Based on such relationships, in our study, we hypothesized that the nasal mucociliary test might help diagnose thyroid function status.

In the current research, we did not specifically measure mitochondrial metabolism. However, understanding the role of mitochondria is an important step toward identifying the plausible pathways of mitochondrial metabolism and nasal ciliary function. The word 'Insights' was deliberately applied in the main title as a critical indicator of our approach. We used the term "Insights" to indicate that our study provides a perspective on the interplay between mitochondrial metabolism, nasal mucociliary clearance, and thyroid dysfunction. This study investigated the possibility of using nasal mucociliary function as a marker for thyroid dysfunction.

We deeply appreciate the Reviewer's suggestion to consider assays or methods to assess the mitochondrial metabolic state in future experiments. This insightful input has the potential to significantly enhance our understanding of the association we are investigating. We eagerly look forward to exploring this direction in our future studies.

Reviewer 2 Report

Comments and Suggestions for Authors

Farhadi and Ghanbari et al. developed a non-invasive method for the efficient diagnosis of complications associated with thyroid dysfunction. Rather than sampling patients' blood, their approach relies on nasal mucociliary (NMC) test following the application of aqueous extract obtained from Iranica Picris of the Asteraceae plant family, conferring a high sensitivity. The results are convincing and therefore this simple but compact study merits the attention of clinicians.

1) Please provide description of how "sensitivity" and "specificity" (Figure 4) were calculated in the Materials and Methods section.

2) Please define abbreviation for "ENT" (line 6), "UP" (line 12), "ROC" (line 29), "AUROC" (line 30), "H" (line 206).

3) Please change "author" to "authors" (line 19).

4) "To explore alternative diagnostic methods to assess thyroid function in patients unable to undergo blood tests for thyroid-stimulating hormone (TSH) and thyroxine (T4), such as individuals with trypanophobia, severe medical conditions, and coagulopathy" (line 20) lacks a verb. Please rephrase.

5) Please replace "Two-hundred thirty-two" with "232" (line 28).

6) Please change "female" to "females" (line 28).

7) Please replace "AUROC=0.82, P=0.004" with "AUROC = 0.82, P = 0.004" (line 30).

8) Please change "AUROC=0.78, P=0.01" to "AUROC = 0.78, P = 0.01" (line 31).

9) Please replace "r=0.47, P<0.001" with "r = 0.47, P < 0.001" (line 32).

10) Please change "P<0.001" to "P < 0.001" (line 33 2x).

11) Please replace "P=0.003" with "P = 0.003" (line 35).

12) Please change "P=0.02" to "P = 0.02" (line 36).

13) "Iranica Picris-based NMC test might serve as a diagnostic method to distinguish overt hypothyroidism and subclinical hyperthyroidism" (line 36) comes rather as a surprise in Conclusions since there seems to be no previous mention of "Iranica Picris" (line 36).

14) When hovering a mouse cursor over the Graphical Abstract, a "A diagram of a human body Description automatically generated" message appears. Please remove this feature.

15) From the legend to the Graphical Abstract is not clear the definition of "total T4". Please fix.

16) Please develop the rationale for "It has been shown that hypothyroidism can lead to mucociliary dysfunction" (line 88) to bring it properly into context.

17) Please replace "of [blinded for review] for" with "for" (line 103).

18) Please change "interfere" to "interfering" (line 117).

19) Please replace "Board of [blinded for review]" with "Board" (line 142).

20) Please change "https://www.strobe-statement.org/checklists/" to "https://www.strobe-statement.org/checklists" (line 146).

21) The link "https://www.strobe-statement.org/checklists/" seems to be dysfunctional (line 146). Please correct.

22) Please provide the name of the "GraphPad Prism 9.5.1" supplier including its city and state affiliation.

23) Please replace "two hundred-one" with "201" (line 173).

24) Please change "minute" to "min" and "Mucociliary" to "mucociliary" in Table 1.

25) When hovering a mouse cursor over Figure 1, a "A diagram of a blue line Description automatically generated with medium confidence" message appears. Please remove this feature.

26) Please replace "Between" with "between" (line 189).

27) Please change "(For interpretation of the correlation coefficient, please refer to the ‘Statistical Analysis’ section)" to "For interpretation of the correlation coefficient, please refer to the ‘Statistical Analysis’ section" (lines 194, 240).

28) Please adjust the font and box size of the graph legends of Figure 2 so that they become equal between panels A and B.

29) When hovering a mouse cursor over Figure 2, a "A diagram of a graph Description automatically generated with medium confidence" message appears. Please remove this feature.

30) Please replace "Over" with "over" (line 203).

31) Please change "3A-C" to "3A–C" (line 208).

32) Please replace "85%" with "85" (line 226).

33) Please change "CI95%" to "CI 95%" (lines 237, 239).

34) Please replace "indicate the" with "indicate" (line 241).

35) Please change "85.7%" to "85.7" (line 249).

36) Please replace "63.7%" with "63.7" (line 250).

37) Please change "1 - Specificity %" to "Specificity %" in the x-axes of the plots shown in Figure 4A–D.

38) Please replace "cells’" with "cell" (line 278).

39) "Recent evidence addressing the direct impact of TSH on mitochondrial metabolism" lacks a verb (line 283). Please revise.

40) Please change "follows. Low" to "low" (line 288).

41) Please replace "reduce its support on" with "reduce" or "prevent" (line 289).

42) Please change "detect" to something like "devise" (line 291).

43) Please replace "represent" with something like "indicate" (line 294).

44) Please change "Ma" to something like "Nevertheless, Ma" (line 295).

45) Please replace "(with elevated TSH levels), oxidative biomarkers (such as gp91phox)" with "with elevated TSH levels, oxidative biomarkers such as gp91phox" (line 297).

46) Please provide reference for "Another study demonstrated that fibulin-1 is associated with thyroid-associated ophthalmopathy" (line 298).

47) Please change "realized" to "discovered" (line 299).

48) The authors mention "The current study's findings" and "this study" in "The current study's findings must be interpreted considering the following limitation: (1) in this study, we included the total T4 values" (line 302), however they refer to "a handful of studies" in "Despite these limitations, this is a handful of studies evaluating the impact of thyroid function on NMC, and the first study introducing the NMC test as a method of thyroid function assessment" (line 308). Consistent with the former sentence, please convert "a handful of studies" to its singular form also in the latter statement.

49) Please replace "current study's findings" with "findings of the current study" (line 302).

50) Please change "in this study, we" to "we" (line 303).

51) Please replace "It is suggested" with something like "We suggest that" (line 303).

52) Please change "was" to "included" (line 304).

53) Please replace "categorize the results per" with something like "stratify with respect to" (line 307).

54) Please change "indicated" to something like "highlights" (line 312).

55) Please replace "hypothyroidism (or subclinical hyperthyroidism)" with "or subclinical hyperthyroidism" (line 312).

56) Please change "distinguish" to "diagnose" (line 314).

57) It is not clear what the authors mean by "traditional" in "Notably, this test may offer a non-invasive alternative for diagnosing thyroid dysfunction in individuals with trypanophobia, or those with coagulopathy or reluctant to undergo traditional for any reason" (line 317). Do they actually mean to say "conventional diagnostic tests"?

58) Please replace "or those" with "those" (line 318).

59) Please change "of" to "for" (line 320).

60) Please replace "for" with "of" (line 321).

61) Please change "F.TH." to "F.T.H." (line 324).

62) Please replace "confirmed the final" with something like "proofread the final version of the" (line 324).

63) Please change "in" to "to" (line 325).

64) It is not exactly clear what the authors refer to by "minor and warning signs" in "Every patient and his associates were educated about minor and warning signs" (line 334)? Do they actually mean to say "minor side effects"?

65) Please replace "study (and their attendants)" with "study and their attendants" (line 339).

66) The link "https://www.calculator.net/sample-size-calculator.html?type=1&cl=95&ci=5&pp=15&ps=&x=121&y=9" seems to be dysfunctional (line 373). Please fix.

Author Response

Dear esteemed Editor and Reviewers

We are submitting the revision of the manuscript. Thank you for your valuable comments, which have improved the quality and clarity of the paper. In the revised manuscript, the responses to the Reviewers’ comments are highlighted in yellow. We tried our best to improve the manuscript per the Reviewers’ comments. I genuinely appreciate the Editor’s and Reviewers’ valuable work and hope the corrections meet the approval requirements.

Best Regards,

Farzad Taghizadeh-Hesary, M.D. 

Esteemed Reviewer 2

Summary and General Evaluation: Farhadi and Ghanbari et al. developed a non-invasive method for the efficient diagnosis of complications associated with thyroid dysfunction. Rather than sampling patients' blood, their approach relies on nasal mucociliary (NMC) test following the application of aqueous extract obtained from Iranica Picris of the Asteraceae plant family, conferring a high sensitivity. The results are convincing and therefore this simple but compact study merits the attention of clinicians.

Response:  I value the positive feedback from the esteemed reviewer. Such comments are motivating and inspire me to pursue further advancements in my experimental studies. I want to reassure the esteemed Reviewer that I have made every effort to address the comments in the revised manuscript.

Comment 1: Please provide description of how "sensitivity" and "specificity" (Figure 4) were calculated in the Materials and Methods section.

Response 1: This comment is appreciated. To clarify this information, the following sentence was added to the study objectives:

"and to calculate the sensitivity and specificity of the NMC test in detecting thyroid dysfunction."

And, the corresponding sentence in the “Statistical Analysis” was revised as follows:

"Sensitivity and specificity values for the Iranica Picris-based NMC test were calculated based on the maximum likelihood ratio (, as determined from the data obtained in the ROC curve analysis."

Comment 2: Please define abbreviation for "ENT" (line 6), "UP" (line 12), "ROC" (line 29), "AUROC" (line 30), "H" (line 206).

Response 2: Thanks for this comment. “ENT” and “UP” are parts of the authors’ affiliations. Therefore, we need to keep them. The full terms of “ROC” and “AUROC” are added to the abstract. There is no full term available for “H” parameter. In the Kruskal-Wallis test, “H” parameter refers to the measure of differences between the medians in the various groups of data using the ranks of data samples in each group.

Comment 3: Please change "author" to "authors" (line 19).

Response 3: Done.

Comment 4: "To explore alternative diagnostic methods to assess thyroid function in patients unable to undergo blood tests for thyroid-stimulating hormone (TSH) and thyroxine (T4), such as individuals with trypanophobia, severe medical conditions, and coagulopathy" (line 20) lacks a verb. Please rephrase.

Response 4: Done. The new sentence is as follows:

"This study aims to explore alternative diagnostic methods to assess thyroid function in patients unable to undergo blood tests for thyroid-stimulating hormone (TSH) and thyroxine (T4), such as individuals with trypanophobia, severe medical conditions, and coagulopathy."

Comment 5: Please replace "Two-hundred thirty-two" with "232" (line 28).

Response 5: Done.

Comment 6: Please change "female" to "females" (line 28).

Response 6: Done.

Comment 7: Please replace "AUROC=0.82, P=0.004" with "AUROC = 0.82, P = 0.004" (line 30).

Response 7: Done.

Comment 8: Please change "AUROC=0.78, P=0.01" to "AUROC = 0.78, P = 0.01" (line 31).

Response 8: Done.

Comment 9: Please replace "r=0.47, P<0.001" with "r = 0.47, P < 0.001" (line 32).

Response 9: Done.

Comment 10: Please change "P<0.001" to "P < 0.001" (line 33 2x).

Response 10: Done.

Comment 11: Please replace "P=0.003" with "P = 0.003" (line 35).

Response 11: Done.

Comment 12: Please change "P=0.02" to "P = 0.02" (line 36).

Response 12: Done.

Comment 13: "Iranica Picris-based NMC test might serve as a diagnostic method to distinguish overt hypothyroidism and subclinical hyperthyroidism" (line 36) comes rather as a surprise in Conclusions since there seems to be no previous mention of "Iranica Picris" (line 36).

Response 13: Thanks in advance for this comment. The abstract was revised as follows:

"The primary endpoint was the diagnostic value of the nasal mucociliary (NMC) test, using Iranica Picris (Asteraceae) aqueous extract, in differentiating hypo- or hyperthyroidism from euthyroid cases."

Comment 14: When hovering a mouse cursor over the Graphical Abstract, a "A diagram of a human body Description automatically generated" message appears. Please remove this feature.

Response 14: We appreciate the detailed review provided by the esteemed Reviewer. We concur with the esteemed reviewer's assessment. We have explored all potential causes of this issue (including original and submitted file name) but have not identified a solution. We are grateful to the esteemed reviewer for his/her understanding.

Comment 15: From the legend to the Graphical Abstract is not clear the definition of "total T4". Please fix.

Response 15: Thanks for this comment. The revised capture is as follows:

"Graphical Abstract. The study summary. IPAE indicates Iranica Picris aqueous extract; NMC, nasal mucociliary; total T4, sum of both bound and free thyroxine; TSH, thyroid-stimulating hormone."

Comment 16: Please develop the rationale for "It has been shown that hypothyroidism can lead to mucociliary dysfunction" (line 88) to bring it properly into context.

Response 16: We appreciate this comment. The following text is added to the corresponding section: (pages 4–5)

"Uysal et al. evaluated the effects of hypothyroidism on nasal mucociliary clearance in patients undergoing total or near total thyroidectomy and planned to receive radioactive iodine. The study demonstrated that mucociliary clearance time was significantly longer during hypothyroid compared to euthyroid periods (16.7 min vs. 9.5 min) [12]."

Comment 17: Please replace "of [blinded for review] for" with "for" (line 103).

Response 17: The corresponding sentence was updated as follows:

"Participants were referred to the Otorhinolaryngology Department of Rasoul Akram Hospital (Tehran, Iran) for purposes other than thyroid care."

Comment 18: Please change "interfere" to "interfering" (line 117).

Response 18: Done.

Comment 19: Please replace "Board of [blinded for review]" with "Board" (line 142).

Response 19: The corresponding sentence was updated as follows:

"The experimental protocols followed the established guidelines and regulations and were approved by the Institutional Review Board of Iran University of Medical Sciences (Tehran, Iran). " 

Comment 20: Please change "https://www.strobe-statement.org/checklists/" to "https://www.strobe-statement.org/checklists" (line 146).

Response 20: Done.

Comment 21: The link "https://www.strobe-statement.org/checklists/" seems to be dysfunctional (line 146). Please correct.

Response 21: Fixed and rechecked. Appreciated!

Comment 22: Please provide the name of the "GraphPad Prism 9.5.1" supplier including its city and state affiliation.

Response 22: Added.

Comment 23: Please replace "two hundred-one" with "201" (line 173).

Response 23: Done.

Comment 24: Please change "minute" to "min" and "Mucociliary" to "mucociliary" in Table 1.

Response 24: Done.

Comment 25: When hovering a mouse cursor over Figure 1, a "A diagram of a blue line Description automatically generated with medium confidence" message appears. Please remove this feature.

Response 25: Thanks for this comment. We concur with the esteemed reviewer's assessment. We have explored all potential causes of this issue (including original and submitted file name) but have not identified a solution. We are grateful to the esteemed reviewer for his/her understanding.

Comment 26: Please replace "Between" with "between" (line 189).

Response 26: Done.

Comment 27: Please change "(For interpretation of the correlation coefficient, please refer to the ‘Statistical Analysis’ section)" to "For interpretation of the correlation coefficient, please refer to the ‘Statistical Analysis’ section" (lines 194, 240).

Response 27: Done.

Comment 28: Please adjust the font and box size of the graph legends of Figure 2 so that they become equal between panels A and B.

Response 28: Thanks for your elaborate review. Figure 2 was revised per this comment.

Comment 29: When hovering a mouse cursor over Figure 2, a "A diagram of a graph Description automatically generated with medium confidence" message appears. Please remove this feature.

Response 29: We concur with the esteemed reviewer's assessment. We have explored all potential causes of this issue (including original and submitted file name) but have not identified a solution. We hope this issue has been resolved in the revised figure.

Comment 30: Please replace "Over" with "over" (line 203).

Response 30: Done.

Comment 31: Please change "3A-C" to "3A–C" (line 208).

Response 31: Done.

Comment 32: Please replace "85%" with "85" (line 226).

Response 32: Done.

Comment 33: Please change "CI95%" to "CI 95%" (lines 237, 239).

Response 33: Done.

Comment 34: Please replace "indicate the" with "indicate" (line 241).

Response 34: Done.

Comment 35: Please change "85.7%" to "85.7" (line 249).

Response 35: Done.

Comment 36: Please replace "63.7%" with "63.7" (line 250).

Response 36: Done.

Comment 37: Please change "1 - Specificity %" to "Specificity %" in the x-axes of the plots shown in Figure 4A–D.

Response 37: Thanks for this comment. According to the standard ROC curve, the x-axis title is “1- Specificity”. Therefore, we need to keep the current format. We appreciate the esteemed Reviewer for his/her understanding.

Comment 38: Please replace "cells’" with "cell" (line 278).

Response 38: Done.

Comment 39: "Recent evidence addressing the direct impact of TSH on mitochondrial metabolism" lacks a verb (line 283). Please revise.

Response 39: Done.

Comment 40: Please change "follows. Low" to "low" (line 288).

Response 40: Done.

Comment 41: Please replace "reduce its support on" with "reduce" or "prevent" (line 289).

Response 41: Done.

Comment 42: Please change "detect" to something like "devise" (line 291).

Response 42: Done, thanks.  

Comment 43: Please replace "represent" with something like "indicate" (line 294).

Response 43: Done.

Comment 44: Please change "Ma" to something like "Nevertheless, Ma" (line 295).

Response 44: Done.

Comment 45: Please replace "(with elevated TSH levels), oxidative biomarkers (such as gp91phox)" with "with elevated TSH levels, oxidative biomarkers such as gp91phox" (line 297).

Response 45: Done.

Comment 46: Please provide reference for "Another study demonstrated that fibulin-1 is associated with thyroid-associated ophthalmopathy" (line 298).

Response 46: Thanks for this comment. The reference [25] was already cited. To clarify it, the sentences were revised as follows:

"It has been demonstrated that fibulin-1 is associated with thyroid-associated ophthalmopathy. In this study, Hu et al. realized that serum fibulin-1 levels higher than 625 pg.mL-1 significantly increased the risk of thyroid-associated ophthalmopathy [25]."

Comment 47: Please change "realized" to "discovered" (line 299).

Response 47: Done.

Comment 48: The authors mention "The current study's findings" and "this study" in "The current study's findings must be interpreted considering the following limitation: (1) in this study, we included the total T4 values" (line 302), however they refer to "a handful of studies" in "Despite these limitations, this is a handful of studies evaluating the impact of thyroid function on NMC, and the first study introducing the NMC test as a method of thyroid function assessment" (line 308). Consistent with the former sentence, please convert "a handful of studies" to its singular form also in the latter statement.

Response 48: Thanks in advance for this comment. The corresponding paragraph was revised as follows:

"The findings of the current study must be interpreted considering the following limitation: (1) we evaluated the total T4 values. We suggest that also consider the free (unbound) T4 in the analysis. (2) The population study includedmainly middle-aged females. Further larger studies are required to explore the results in males and at the extremes of age. (3) The impacts of tobacco smoking on TSH and T4 levels have been demonstrated [26]. The current study did not stratify the results with respect to tobacco smoking status. Despite these limitations, this is among a handful of studies evaluating the impact of thyroid function on NMC, and the first study introducing the NMC test as a method of thyroid function assessment."

Comment 49: Please replace "current study's findings" with "findings of the current study" (line 302).

Response 49: Done.

Comment 50: Please change "in this study, we" to "we" (line 303).

Response 50: Done.

Comment 51: Please replace "It is suggested" with something like "We suggest that" (line 303).

Response 51: Done.

Comment 52: Please change "was" to "included" (line 304).

Response 52: Done.

Comment 53: Please replace "categorize the results per" with something like "stratify with respect to" (line 307).

Response 53: Done.

Comment 54: Please change "indicated" to something like "highlights" (line 312).

Response 54: Done.

Comment 55: Please replace "hypothyroidism (or subclinical hyperthyroidism)" with "or subclinical hyperthyroidism" (line 312).

Response 55: Done.

Comment 56: Please change "distinguish" to "diagnose" (line 314).

Response 56: Done.

Comment 57: It is not clear what the authors mean by "traditional" in "Notably, this test may offer a non-invasive alternative for diagnosing thyroid dysfunction in individuals with trypanophobia, or those with coagulopathy or reluctant to undergo traditional for any reason" (line 317). Do they actually mean to say "conventional diagnostic tests"?

Response 57: Thanks in advance for this comment. The sentence was revised as follows:

"Notably, this test may offer a non-invasive alternative for diagnosing thyroid dysfunction in individuals with trypanophobia, or those with coagulopathy or reluctant to undergo conventional diagnostic tests for any reason."

Comment 58: Please replace "or those" with "those" (line 318).

Response 58: Done.

Comment 59: Please change "of" to "for" (line 320).

Response 59: Done.

Comment 60: Please replace "for" with "of" (line 321).

Response 60: Done.

Comment 61: Please change "F.TH." to "F.T.H." (line 324).

Response 61: Done.

Comment 62: Please replace "confirmed the final" with something like "proofread the final version of the" (line 324).

Response 62: Done.

Comment 63: Please change "in" to "to" (line 325).

Response 63: Done.

Comment 64: It is not exactly clear what the authors refer to by "minor and warning signs" in "Every patient and his associates were educated about minor and warning signs" (line 334)? Do they actually mean to say "minor side effects"?

Response 64: Corrected. Thanks.

 Comment 65: Please replace "study (and their attendants)" with "study and their attendants" (line 339).

Response 65: Done.

Comment 66: The link "https://www.calculator.net/sample-size-calculator.html?type=1&cl=95&ci=5&pp=15&ps=&x=121&y=9" seems to be dysfunctional (line 373). Please fix.

Response 66: Fixed and rechecked. Appreciated!  

Round 2

Reviewer 1 Report

Comments and Suggestions for Authors

This reviewer still does not understand that how authors connect Mitochondrial metabolism as a diagnostic method for thyroid dysfunction without performing any study specifically on mitochondria or related metabolic pathways. Authors argued that mitochondrial function is inherently linked to NMF but this is true for almost all cellular and organ functions as mitochondria is the major source of cellular energy and metabolites.  

Author Response

Esteemed Reviewer 1

Comment: This reviewer still does not understand that how authors connect Mitochondrial metabolism as a diagnostic method for thyroid dysfunction without performing any study specifically on mitochondria or related metabolic pathways. Authors argued that mitochondrial function is inherently linked to NMF but this is true for almost all cellular and organ functions as mitochondria is the major source of cellular energy and metabolites.  

Reply: We appreciate the esteemed reviewer’s comment. It guided us to search the literature to explore more mechanisms linking thyroid hormones–mitochondria–cilia nexus. 

The following paragraph was added to the revised manuscript to clarify this association (lines 283–291):

"Cilia beating is an ATP-dependent process. Kinesin family member 19A (Kif19a) is a motor protein that attaches to the cilia tubulin in an ATP-dependent manner [21]. Therefore, normal cilia dynamics depend on functional mitochondria generating enough ATP molecules [22]. Furthermore, elevated levels of extracellular ATP have been shown to enhance ciliary beating in ciliated epithelial cells by inducing membrane hyperpolarization through a calcium-dependent process [23], [24]. Research suggests that mitochondria are the primary supplier of extracellular ATP in the vicinity of epithelial cells [25]. This paragraph demonstrated how mitochondrial metabolism directly affects cilia dynamics by enhancing intra- and extracellular ATP contents."

We appreciate the esteemed reviewer’s comment. It guided us to search the literature to explore more mechanisms linking thyroid hormones–mitochondria–cilia nexus. 

It is important to acknowledge that the connection between mitochondrial function and thyroid hormones in the context of the nasal mucociliary test (NMC test) is based on established scientific evidence and known physiological mechanisms. The hypothesis proposed in our study suggests that the NMC test could serve as an alternative diagnostic method for evaluating thyroid functional status, potentially through its interaction with mitochondria. 

The rationale behind this hypothesis lies in the existing literature supporting the following key points: 

  1. Thyroid hormones have a known stimulating effect on nasal ciliary function, as evidenced by studies showing reduced nasal ciliary function in patients with hypothyroidism (PMID: 23519683, PMID: 32754873).
  2. Thyroid hormones play a direct role in activating mitochondrial metabolism in target cells. They stimulate mitochondrial replication and energy production within mitochondria by switching metabolism from glycolytic pathway to more efficient oxidative phosphorylation (PMID: 35326451, PMID: 31778245).

  3. Ciliary beating, a crucial aspect of nasal mucociliary clearance, is an ATP-dependent process that relies on proper mitochondrial metabolism. Kif19a is a motor protein that attaches to tubulin in an energy-dependent manner using ATP (PMID: 23168168). Moreover, increase in extracellular ATP can stimulate cilia beating placed on ciliated epithelial cells by hyperpolarizing the plasma membrane through a calcium-dependent mechanism (PMID: 7756536, PMID: 28422293). Evidence supports that mitochondria are the main source of extracellular ATP around epithelial cells (PMID: 12948585). 

Overall, four mechanisms (2 direct, 2 indirect) have been put forward in the revised manuscript (lines 283–305) postulating how mitochondrial metabolism can potentially mediate normal ciliary dynamics. 

By integrating these established facts, we posit that the NMC test, which evaluates nasal ciliary function, could indirectly reflect thyroid hormonal status through its reliance on ATP-dependent ciliary beating and mitochondrial function. While it is true that mitochondria are vital for various cellular functions beyond NMC, the specific link between thyroid hormones, mitochondrial metabolism, and nasal ciliary function forms the basis of our innovative approach to exploring thyroid functional status through NMC test.

We recognize the importance of further research specifically targeting mitochondria and related metabolic pathways to validate this hypothesis comprehensively. The existing evidence and theoretical framework presented in our study offer a solid foundation for investigating the potential utility of the NMC test in assessing thyroid function, highlighting a promising avenue for future research in this field. In our upcoming research, we aim to delve deeper into unraveling the molecular mechanisms that underlie this intricate interplay. We appreciate the reviewer for their invaluable feedback and their dedication to enhancing the quality of this manuscript

Round 3

Reviewer 1 Report

Comments and Suggestions for Authors

Manuscript may be accepted.